# Integration of Metal-Organic Frameworks with Bi-Nanoprobes as Dual-Emissive Ratiometric Sensors for Fast and Highly Sensitive Determination of Food Hazards

**DOI:** 10.3390/molecules27072356

**Published:** 2022-04-06

**Authors:** Chi-Xuan Yao, Lu Dong, Lu Yang, Jin Wang, Shi-Jie Li, Huan Lv, Xue-Meng Ji, Jing-Min Liu, Shuo Wang

**Affiliations:** Tianjin Key Laboratory of Food Science and Health, School of Medicine, Nankai University, Tianjin 300071, China; mthgh1986@163.com (C.-X.Y.); donglu@nankai.edu.cn (L.D.); yl1215924627@163.com (L.Y.); wangjin@nankai.edu.cn (J.W.); lsj930427@126.com (S.-J.L.); lvhuan@nankai.edu.cn (H.L.); jixuemeng@nankai.edu.cn (X.-M.J.)

**Keywords:** ZIF-67, heavy metal ions, food-safety, ratiometric fluorescence

## Abstract

Functional nanoprobes which detect specific food hazards quickly and simply are still in high demand in the field of food-safety inspection research. In the present work, a dual-emission metal-organic framework-based ratiometric fluorescence probe was integrated to detect Cu^2+^ and Pb^2+^ with rapidness and ease. Specifically, quantum dots (QDs) and carbon quantum dots (CQDs) were successfully embedded into zeolitic imidazolate framework-67 (ZIF-67) to function as a novel ratiometric fluorescent sensing composite. The ratiometric fluorescence signal of CQDs/QDs@ZIF-67 was significantly aligned with the concentration of metal ions to give an extremely low detection limit of 0.3324 nM. The highly sensitive and selective CQDs/QDs@ZIF-67 composite showed potential for the rapid and cost-effective detection of two metal ions.

## 1. Introduction

Food safety and environmental contamination are two of the most important matters in contemporary society. Exposure to water contaminants, such as toxic anions and heavy metal ions, is a worldwide problem owing to its detrimental impacts on the environment, food security, and human health [1,2]. Heavy metal ions are important factors in water pollution due to their wide applications in many domains, including the mining sector, the chemical industry, the manufacture of batteries, and other industries [3,4,5]. Moreover, heavy metal ions can participate in receptor binding and accumulate in organs, causing serious diseases and organ failure after they enter the body [6,7]. Among them, copper ions (Cu^2+^) are necessary for the physiological activities of animals and plants [8,9]. However, excess Cu^2+^ can lead to kidney or liver damage and gastrointestinal disorders, disrupt cellular metabolism, etc. [10,11,12]. With the swift growth of industrial effluent and acid-mine water, Cu^2+^ contamination of food, drinking water, and various industrial products is becoming an increasingly serious concern for contemporary society [13,14,15,16,17]. Furthermore, persistent lead (Pb^2+^) pollution has aroused widespread public concerns because it has been condemned as a major threat to human health, due to its exposure through numerous pathways globally. Lead exposure mainly arises through the respiratory tract, or through contact with contaminated hands, food, water, cigarettes, or clothing. As such, Pb^2+^ is a major environmental pollutant; trace Pb^2+^ can cause irreversible damage to the brain and the central nervous system [18].

Metal-organic frameworks (MOFs) have gained extensive attention because of their excellent performance. MOFs can be used as platforms for the design and fabrication of multifunctional chemo-sensors and biosensors, due to their unique porousness, adaptable structures or compositions, and good permanence [19,20,21,22]. MOFs and their derivatives show stable luminescence, chemical usefulness, and strong biocompatibility towards sensing probes such as antibodies or aptamers; they also display strong abilities as emitters, or biosensors carriers for the selective and sensitive detection of deleterious substances in food products. By combining different detection techniques such as fluorescence [23], electrochemical detection [24], or flame atomic absorption spectrometry [25], MOF-based synthetic materials have demonstrated the potential to be used for monitoring heavy metals. Additionally, MOF-based synthetic materials are increasingly used for the analysis of food contaminants; these recent developments have been discussed [26].

Since the sensitive and selective detection of Cu^2+^ and Pb^2+^ is of vital importance to human health, there is a pressing need to develop a novel sensing material to rapidly identify and detect metal ions. Some traditional analytical techniques are struggling to satisfy the demands of the real-time detection because their processes are time consuming and complicated [27,28,29]. With the advancement of optical sensing research, ratiometric fluorescent sensor technology has demonstrated remarkable properties, such as a low cost, simple operation, high sensitivity, and short response time. It is critical to design a fluorescent material when using fluorescent sensors in practice. Despite the great sensitivity, precision, and dependability of standard methods for pollutants analysis, these conventional strategies also involve complicated apparatus, well-trained personalized operation, and laborious and time-consuming procedures.

With these insights, we propose a new kind of fluorescent composite for the rapid detection of Cu^2+^ and Pb^2+^ at room temperature. This composite (named CQDs/QDs@ZIF-67), which emits dual fluorescent peaks based on one-pot synthesis, is composed of quantum dots (QDs), carbon quantum dots (CQDs), and MOFs. The blue fluorescent CQDs were used for the internal standard material, and the red fluorescent QDs were used for the response signal. The fluorescence intensity of the CQDs was in proportion with the fluorescence intensity of the QDs so that background interference could be effectively avoided in the quantitative analysis. Cu^2+^ and Pb^2+^ can both cause the fluorescence quenching of QDs due to the high binding of QDs and the sensitization of MOFs, and then induce fluorescence color change of the CQDs/QDs@ZIF-67 suspension from pink to purple. CQD/QDs@ZIF-67 exhibited novelty and superiority as an optical probe in terms of sensitivity, ease of operation, and rapidness, as well as robustness, thus promising to be a new portable detector for rapid food-safety inspection and screening.

## 2. Results and Discussion

### 2.1. Preparation of the CQDs/QDs@ZIF-67 Composite

Zeolitic imidazolate frameworks (ZIFs), as a subclass of MOFs, combine the advantages of MOFs and zeolitesmaterials; in particular, ZIFs possess high thermal and chemical stability [30,31,32]. Recently, zeolitic imidazolate framework-67 (ZIF-67) has been effectively used as a sorbent to remove organic pollutants from water; combined with its impressive hydrothermal stability, this presents its potential for water purification [33,34,35]. However, there are some emerging reports on the adsorption of heavy metal ions by ZIF-67 materials. In this paper, both the adsorption of single heavy metal ions and the competitive adsorption characteristics of hazardous metal species on ZIF-67 crystals were investigated. They can be efficiently adsorbed by ZIF-67 due to the coordination with the pyridine nitrogen-containing ligands of 2-Methylimidazole [36,37]. Therefore, ZIF-67 was chosen as the supporter to carry fluorescence nanomaterials to establish novel composites for the ultrasensitive detection of heavy metal ions (Figure 1).

In this work, the CQDs/QDs@ZIF-67 composite was prepared by a one-pot synthesis with orderly crystalline structures. The fluorescence nanomaterials could be directly incorporated into the synthesis of ZIF-67 to form a class of hybrid metal-organic framework materials (Figure 1). The as-prepared CQDs/QDs@ZIF-67 composite dispersed in water appeared milky purple in sunlight (left inset in Figure 2A) and emitted pink fluorescence (right inset in Figure 2A) in ultraviolet light (365 nm). As shown in Figure 2A, the composite and the mixture of CQDs and QDs had almost identical fluorescence spectra, with emission peaks at 425 nm and 630 nm upon excitation at 360 nm. Additionally, pure ZIF-67 dispersion showed no visible fluorescence emission peak, which suggests that the pink fluorescence of the composite originated from the CQDs and QDs rather than from the ZIF-67. Moreover, the ZIF-67 shell had almost no effect on the fluorescence of CQDs and QDs. As shown in Figure 2B, under different excitation wavelengths, the fluorescence-emitting position of CQDs/QDs@ZIF-67 can be kept almost unchanged at 630 nm, which belongs to the characteristic emission peak of QDs. On the contrary, the fluorescence emission peak of CQDs varies with excitation, and they have a maximum emission peak with the excited wavelength at 350 nm. For the above-mentioned phenomenon, 360 nm was selected as the detection excitation wavelength due to the remarkable intensity of QDs in CQDs/QDs@ZIF-67.

In order to further confirm that the fluorescence nanomaterials (CQDs and QDs) were doped into ZIF-67 during the synthesis process, rather than adsorbed on the surface of ZIF-67, the prepared CQDs/QDs@ZIF-67 composites were washed with plenty of water five times and their fluorescence spectra were recorded. Compared with the simple mixing of QDs, CQDs and ZIF-67, it was observed that the fluorescence of the mixture decreased significantly while CQDs/QDs@ZIF-67 still maintained strong fluorescence (Figure 2E). Therefore, QDs and CQDs were embedded during the synthesis of the MOFs, rather than adsorbed physically on the surfaces of the MOFs. 

At the same time, the stability of CQDs/QDs@ZIF-67 was also investigated in this paper. The CQDs/QDs@ZIF-67 composite had good fluorescence stability and good fluorescence performance within 25 days (Figure 2F).

As shown in Figure 2C, the PXRD curves reveal that pure ZIF-67 was provided with the cubic sodalite-related crystal structures, which coincided with that of the reported ZIF-67 formation, indicating the successful synthesis of ZIF-67. Additionally, the CQDs/QDs@ZIF-67 composite had diffractive peaks in PXRD, as did the pristine ZIF-67, which fully confirmed that the assembling of CQDs and QDs had no adverse impact on the crystal perfection of ZIF-67.

The FT-IR spectra of CQDs, QDs, ZIF-67, and CQDs/QDs@ZIF-67 are shown in Figure 2D. It can be seen that the spectrum of the CQDs/QDs@ZIF-67 composite had stretch bands at 2933.6 cm*^−^*^1^, 1575.0 cm*^−^*^1^, 1386.6 cm*^−^*^1^ and 995.6 cm*^−^*^1^, which were homologous with the spectra of CQDs and QDs, indicating the successful merging of the fluorescence materials and ZIF-67 for the production of the CQDs/QDs@ZIF-67 composite.

The TEM images of QDs and CQDs are also shown in Figure 3A,B. The average sizes of the QDs and CQDs were 11.4 nm and 2.23 nm, respectively. The TEM images show that the synthesized composite exhibited a nearly identical morphology to the pure ZIF-67 crystalline shown in the previous paper, with a size of approximately 460 nm (Figure 3C,D) [38].

### 2.2. The Sensing Performance of CQDs/QDs@ZIF-67

The selectivity of CQDs/QDs@ZIF-67 was explored in this work (Figure 4A). In the presence of Cu^2+^ (300 nM) or Pb_2+_ (100 μM), the effect of interferents (including Na^+^, Co^2+^, Ca^2+^, Mn^2+^, Al^3+^, Fe^3+^, Zn^2+^, Mg^2+^, Ag^+^, Hg^2+^, and common anions, at 200 μM) on the fluorescence response (F_630_/F_425_) of the CQDs/QDs@ZIF-67 composite was investigated. Figure 4A shows that the fluorescence intensity of CQDs/QDs@ZIF-67 was obviously quenched by Cu^2+^ and Pb^2+^. To expand the application of CQDs/QDs@ZIF-67, many kinds of heavy metal ions were chosen to evaluate the sensing performance of the composite. Other interferents had inferior influences on the fluorescence of QDs/CDs@ZIF-67 composite compared to those of Cu^2+^ and Pb^2+^. The quenching concentration of Cu ions was lower than that of Pb^2+^. The presence of Cu ions influenced Pb detection due to its sensitive detection in the nanomolar range. These results indicate the good sensing abilities of CQDs/QDs@ZIF-67 for the specific recognition of Cu^2+^ and Pb^2+^ in the aqueous environment.

Subsequently, the detection variables, including pH value and reaction time, were optimized for sensing and detecting heavy metal ions. The quenching efficiency (ΔF) was used to judge performance with the equation:ΔF = F^0^_630_/F^0^_425_ − F_630_/F_425_(1)
where F^0^_630_/F^0^_425_ and F_630_/F_425_ are the ratios of fluorescence intensity at 630 nm to 425 nm in the absence and presence of Cu^2+^ (300 nM), separately. The optimized incubation time of CQDs/QDs@ZIF-67 with Cu^2+^ was 15 min (Figure 4B). The best fluorescence stability conditions and the optimized conditions of CQDs/QDs@ZIF-67 were as follows: the pH value of 7 was selected as the optimal pH (Figure 4C,D). 

### 2.3. The Interaction between CQDs/QDs@ZIF-67 and Targets

The fluorescence interaction between CQDs/QDs@ZIF-67 and the two targets is discussed in this section. Cu^2+^ and Pb^2+^ were chosen due to the high quenching effect of Cu^2+^ and Pb^2+^ on the fluorescence of QDs, as well as the well-known interaction between the targets and the imidazole in ZIF-67, which enhances the quenching effect [37]. Figure 5A shows the effect of Cu^2+^ and Pb^2+^ on the fluorescence of CQDs/QDs@ZIF-67. Both Cu^2+^ and Pb^2+^ can quench the fluorescence peak of CQDs/QDs@ZIF-67 at 630 nm, while the fluorescence intensity of CQDs/QDs@ZIF-67 is almost unchanged at 425 nm. This is mainly because Cu^2+^ and Pb^2+^ have strong effects on the photoluminescence properties of QDs. With the addition of Cu^2+^ (300 nM) or Pb^2+^ (100 μM) into the CQDs/QDs@ZIF-67 suspension, the quenching efficiency of CQDs/QDs@ZIF-67 achieved up to 90% of the fluorescence emission peak at 630 nm, while it retained about 60% of the emission intensity of the mixture of CQDs and QDs at 630 nm (Figure 5A). ZIF-67 is a kind of porous material which can easily enrich a variety of heavy metal ions. Based on the enrichment effect of ZIF-67, it promotes increased access to QDs with Cu^2+^ and Pb^2+^, resulting in the improvement of the quenching efficiency of Cu^2+^ [39]. Therefore, the results (in Figure 5A) further demonstrate that the CQDs/QDs@ZIF-67 composite had a much stronger sensing capability than the mixture of CQDs and QDs for the detection of heavy metal ions.

The quenching mechanism of the fluorescence of CQDs/QDs@ZIF-67 was investigated. As shown in Figure 5B, Cu^2+^ and Pb^2+^ had no obvious absorption at wavelengths of 300–600 nm, indicating that the fluorescence-quenching effects between the CQDs/QDs@ZIF-67 composite and target ions were not caused by fluorescence resonance energy transfer or an inner filter effect. We think that the quenching mechanisms of Cu^2+^ and Pb^2+^ on the CQDs/QDs@ZIF-67 composite may be analogous to that of the quenching of CdSe/ZnS QDs by Cu^2+^ and Pb^2+^, which has been studied extensively before [40,41,42,43,44].

The observed quenching phenomena could be due to the cation exchanging of Zn with Pb and Cu [40]. The driving force for exchanges between two cations can be controlled by the difference in electronegativities, bond energies of S-M, formation energies of the sulfide composites, and lattice energies, as well as the solvation energies in the presence of a particular coordinating species. From the observed bond energies, the following sequence as ZnS < CuS < PbS. Accordingly, the dissociation of the Zn–S bond should be much easier than those of Cu–S and Pb–S. Therefore, the transformation of ZnS to PbS and CuS was favored; with the help of the cation exchange reaction of Zn^2+^ with Pb^2+^ and Cu^2+^ [40], Pb will preferentially bind to S, and the band gap width of PbS is narrower than ZnS, so that the binding of the ZnS shell to the carriers was smashed, which results in fluorescence quenching.

The CdSe/ZnS QDs used in the work possessed a core–shell structure capped with an imperfect shell, whereby small cations can pass through the shell and then interact with the QDs’ cores. Therefore, the strong binding of Cu^2+^ onto the surface of the QD cores leads to the fluorescence quenching of the QDs, which further leads to the replacement of surface Cd by Cu, thus forming CuSe particles on the QDs’ boundary [40,45]. The interaction may result in the effective fluorescence quenching of the CQDs/QDs@ZIF-67 composite.

### 2.4. Method Establishment

As shown in Figure 6, with the addition of different concentrations of Cu^2+^ or Pb^2+^, the fluorescence strength of the QDs/CDs@ZIF-67 composite at 425 nm had barely changed, while the fluorescence intensity at 630 nm was obviously quenched. The value (y = F_630_/F_430_) showed a linear increase with the increase in Cu^2+^ concentration in the range of 1–280 nM using the equation: y = −0.0063x + 2.4968(2)
where y is the ratio of fluorescence intensity from 630 nm to 425 nm in the presence of Cu^2+^. The limit of detection (3 s/K, where s is the standard deviation of blank measurements of the composite, and K represents the slope of the calibration graph) was calculated as 0.51 nM. For Pb^2+^, the value (y’ = F’_630_/F’_430_) showed a linear increase with the increase in Pb^2+^ concentration in the range of 2–95 μM using the equation: y’ = −0.024x + 2.6899(3)
where y’ was the ratio of fluorescence intensity at 630 nm to 425 nm in the presence of Pb^2+^. The limit of detection was calculated to be 0.51 nM.

The performance of the designed ratiometric fluorescence sensor based on the CQDs/QDs@ZIF-67 composite was superior to some other Cu^2+^/Pb^2+^ sensors (Table 1). Its sensing performance was likely due to the uniform structures of the CQDs/QDs@ZIF-67 composite, which resulted in a large number of CQDs/QDs@ZIF-67 composites reacting with Cu^2+^ and Pb^2+^ [46,47,48,49].

### 2.5. Application to Practical Samples

The developed composite was applied to the determination of two targets in two types of real samples. Since neither of them was detected in the samples, Cu^2+^ and Pb^2+^ were added to the real samples before testing. The application performance was evaluated from the actual samples, which were adequately diluted with PBS buffer and spiked with increasing amounts of the standard target solution. Accordingly, the heavy metal ions of the samples diluted with PBS buffer were quantified and input into a calibration plot constructed with buffered target standards. 

With a ratio fluorescence probe, all experiments were repeated three times in parallel, and the relative standard deviation was as low as 1.3% (Table 2). The recovery of spiked samples ranged from 94.2% to 101%, which confirms that the prepared CQDS/QDs@ZIF-67 composite has great feasibility in ratiometric fluorescence sensing for Cu^2+^ and Pb^2+^.

## 3. Materials and Methods

### 3.1. Materials

Anhydrous citric acid and 3-Aminopropyltriethoxysilane (APTES) were purchased from Sinopharm Chemical Reagent Co., Ltd. (Shanghai, China). Cobaltous nitrate hexahydrate (CoH_12_N_2_O_12_) and 2-methylimidazole (2-Hmin) were purchased from Sigma-Aldrich Co., Ltd. (Millipore, Burlington, MA, USA). The ionic standard solution was purchased from BeNa Culture Collection (China), and water-soluble CdSe/ZnS quantum dots were purchased from Xingzi New Material Technology Co. Ltd. (Shanghai, China). Other chemical reagents used in this work were purchased from Sinopharm Co. Ltd. (Shanghai, China). Ultrapure water was obtained from a PURELAB Classic water purification system (Millipore, USA). All reagents were of analytical grade and used without any further purification.

### 3.2. Instrumentation

The images of transmission electron microscopy (TEM) were recorded on a JEM-2100 (HR) transmission electron microscope (JEOL, Tokyo, Japan). The images of scanning electron microscopy (SEM) were recorded on a S4800 transmission electron microscope (JEOL, Japan). FT-IR spectra were recorded on a Spectrum 100 spectrometer (Perkin Elmer, Waltham, MA, USA). Powder X-ray diffraction (PXRD) patterns were recorded on a Bruker D8 Advance X-ray diffractometer using Cu Kα radiation at 40 mA and 40 kV. UV-vis absorption and fluorescence spectra were recorded using an Evolution 350 UV-vis spectrophotometer (Thermo, USA) and an F-2500 fluorescence spectrometer (Hitachi, Tokyo, Japan), respectively.

### 3.3. Preparation of Carbon Quantum Dots (CQDs)

The CQDs were created using a one-step hydrothermal method based on previous research [50]. Herein, anhydrous citric acid (carbon source) was dissolved in ultrapure water (26.0 mL) and deaerated with nitrogen for 15 min. The functionalized reagent, APTES (4.0 mL), was then injected. The obtained mixture was stirred for 10 min, and heated at 200 °C for 2 h. The prepared CQDs were purified with a dialysis bag (1 kDa) and stored at 4 °C.

### 3.4. Synthesis of the CQDs/QDs@ZIF-67 Composite

The CQDs/QDs@ZIF-67 composite was prepared by a one-pot method according to previous research with some modification. Firstly, a solution of 2-methylimidazole (0.098 g) in 10 mL of methanol was mixed with 10 mL methanol which contained QDs (0.5 mg mL^−^^1^) and CQDs (1.12 mg mL^−^^1^), and then rapidly poured into 30 mL of 2-methylimidazole (0.098 g) methanol solution. Then, Co(NO_3_)_2_∙6H_2_O (0.058 g) in 10 mL of methanol was immediately added to the above mixture and allowed to react at room temperature for 24 h with stirring. Finally, the CQDs/QDs@ZIF-67 composite was collected by centrifugation (10,000 rpm, 5 min). The obtained composite was washed several times with methanol, and vacuum-dried overnight (60 °C) for further use [51,52].

### 3.5. Determination of Cu^2+^ and Pb^2+^

For typical detection, the CQDs/QDs@ZIF-67 composite suspension (20 mg mL^−^^1^) and Cu^2+^ solutions were prepared with a buffer solution (pH 8). Firstly, 100 µL of the CQDs/QDs@ZIF-67 composite suspension was diluted to 800 µL, and then 100 µL of Cu^2+^/Pb^2+^ standard solution with different concentrations was added into the suspension. Then, the mixtures were vortexed for 45 s and then incubated for 10 min at room temperature, and the fluorescence spectra were recorded upon excitation at 360 nm. The fluorescence-quenching efficiency was calculated according to (F_630_/F_430_), where F_630_ and F_430_ are the fluorescence intensities of CQDs/QDs@ZIF-67 at 630 nm and 425 nm, respectively.

### 3.6. Sample Preparation

The tap water (collected from the lab) and apple juice (purchased from the local market) were diluted 2 times with PBS solution (pH 7) and the pH was adjusted before use. The determination of Cu^2+^ or Pb^2+^ in real samples was performed as described in Section 3.5.

## 4. Conclusions

In this work, CQDs/QDs@ZIF-67 with double fluorescence emission peaks was developed through a one-pot synthesis, in which the CQDs and QDs were doped during the synthesis of ZIF-67. It was validated with a test that the synthesis process had little effect on the fluorescence nanomaterials, and it retained the characteristics of MOFs. CQDs/QDs@ZIF-67 was used as a ratio fluorescence probe because background interference could be effectively avoided, and the effects of heavy metal ions on CQDs/QDs@ZIF-67 were further investigated. It was found that the fluorescence peak of CQDs/QDs@ZIF-67 at 630 nm can be quenched by Cu^2+^ and Pb^2+^ effectively, whereas the blue-emitting CQDs are not quenched. The detection conditions were further optimized in this work. Compared with previous works, CQDs/QDs@ZIF-67 has a good ability to sense Cu^2+^ and Pb^2+^. The developed composite was successfully applied to the determination in real samples. We believe that the use of this MOF-based composite may extend to the sensing of other heavy metals, and even to other harmful substances or food hazards.

## Data Availability

Not applicable.

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
