# Peer review of "Integration of Metal-Organic Frameworks with Bi-Nanoprobes as Dual-Emissive Ratiometric Sensors for Fast and Highly Sensitive Determination of Food Hazards"

_molecules, 2022, doi:10.3390/molecules27072356_

Round 1

Reviewer 1 Report

This work presented a ratiometric sensor based on composite of MOF and two fluorescent quantum dots. The sensing performance is commendable, however several issues need to be addressed before publication.

1. My main question with this work is that the advantage of a ratiometric sensor over a single-probe fluorescent one is not well established. A common ratiometric would benefit from significantly enhanced sensitivity and selectivity due to the opposite trends of two fluorescent probes. What do the CQDs bring to the sensing performance, which does not change significantly or even rationally upon interaction with the metal ions?

The logic gate part is really confusing. If either ion can quench the fluorescence, I do not see much benefit in such a “logic gate”, apart from the low distinguishability between these two ions.

2. The English writing is mostly concise and accurate. Still, some sentences could be hard to comprehend and require polishing. E.g,, “For the above-mentioned phenomenon, the emission peak of QDs required strong enough for detection and CQDs also has the complete emission peak so that 360 nm is selected as the detection excitation wavelength.

3. The washing experiment performed in the work can only prove that QDs are not physically adsorbed on the surface of the MOF crystals. However, it could not rule out the possibility of strong chemical adsorption.

4. The pH dependence of the fluorescence of the ZIF QDs composite should be characterized before testing the pH dependence in ion sensing.

Author Response

The manuscript has been carefully revised in light of the reviewers’ suggestions. Please find the “Revised Manuscript”, and “Response to Reviewers’ Comments”.

Reviewer 2 Report

In the manuscript "Integration of metal-organic frameworks with bi-nanoprobes as dual-emissive ratiometric sensors for fast and highly sensitive determination of food hazards with construction of molecular logic gate" authors propose the method for heavy metal detection in food sample. Authors declare, that this method based on the possibility of MOF's to adsorb the heavy metal ions. Together with the QDs these structures provide spectral analytical signal. The MOF's decorated with QDs demonstrate selective detection of Pb and Cu ions. The proposed sensor approved possibility of Pb and Cu detection in juice samples. Th article recommended to publish after minor revision.

The comments are below:

  1. On p. 2 line 89 authors declare about possibility of heavy metal ions adsorption on the ZIF-67 surface. But there is no links to approve this statement. It is better to clarify the mechanisms of heavy metal interaction with ZIF-67.
  2. The two wavelength serve as analytical signal. But on figures 2a and 2b peaks at these wavelength are not clearly depicted. It is better to show all peaks on these figures.
  3. Need to explain how the presence of Cu ions influence on Pb detection and how the Pb presence influence on Cu detection.
  4. The logic gate system should be clearly described. How it help to detect Cu and Pb together in one sample?
  5. The conclusions should be reconsidered in accordance with all remarks.

Author Response

(The authors gave the same response as above.)

Round 2

Reviewer 1 Report

The manuscript should be alright to publish on Molecules now.